# Layers of the monkey visual cortex are selectively modulated during electrical stimulation

Sangjun Lee[1]*, Zhihe Zhao[1], Ivan Alekseichuk[1,2], Jimin Park[1], Sina Shirinpour[1], Gary Linn[3,4], Charles E. Schroeder[3,5], Arnaud Y. Falchier[3,4], Alexander Opitz[1]*

**1** Department of Biomedical Engineering, University of Minnesota, Minneapolis, Minnesota, United States of America, **2** Stephen M. Stahl Center for Psychiatric Neuroscience, Department of Psychiatry and Behavioral Sciences, Northwestern University Feinberg School of Medicine, Chicago, Illinois, United States of America, **3** Translational Neuroscience Lab Division, Center for Biomedical Imaging and Neuromodulation, The Nathan S. Kline Institute for Psychiatric Research, Orangeburg, New York, United States of America, **4** Department of Psychiatry, NYU Grossman School of Medicine, New York, New York, United States of America, **5** Departments of Neurological Surgery and Psychiatry, Columbia University College of Physicians and Surgeons, New York, New York, United States of America

* lee03936@umn.edu (SL); aopitz@umn.edu (AO)

## Abstract

The mammalian neocortex, organized into six cellular layers or laminae, forms a cortical network within layers. Layer-specific computations are crucial for sensory processing of visual stimuli within the primary visual cortex. Laminar recordings of local field potentials (LFPs) are a powerful tool to study neural activity within cortical layers. Electric brain stimulation is widely used in basic neuroscience and in a large range of clinical applications. However, the layer-specific effects of electric stimulation on LFPs remain unclear. To address this gap, we recorded laminar LFP from capuchin monkeys' primary visual cortex while presenting a flash visual stimulus. Simultaneously, we applied a low-frequency sinusoidal current to the occipital lobe with an offset frequency to the flash stimulus repetition rate. We analyzed the modulation of visual-evoked potentials with respect to the phase of applied electric stimulation. Our results reveal that only the deeper layers, but not the superficial layers, show phase-dependent changes in LFP components with respect to the applied current. Employing a cortical column model, we show that these in vivo observations can be explained by phase-dependent changes in the driving force within neurons of deeper layers. Our findings offer crucial insight into the selective modulation of cortical layers through electrical stimulation, thus advancing approaches for more targeted neuromodulation.

## Introduction

Electrical stimulation has emerged as a promising therapeutic application for treating various neurological and psychiatric symptoms such as depression, epilepsy,

**Data availability statement:** All relevant data are within the paper and its Supporting information files. All data and code files are also available from the Zenodo repository (https://doi.org/10.5281/zenodo.15723993).

**Funding:** This research was supported in part by the National Institute of Health (RF1MH124909 and R01EB034143 to AO, https://www.nimh.nih.gov) and the University of Minnesota's MnDRIVE Initiative (to SL, https://mndrivebrainconditions.umn.edu). The funders had no role in study design, data collection and analysis, decision to publish, or preparation of the manuscript.

**Competing interests:** The authors have declared that no competing interests exist.

**Abbreviations:** AC, alternating current; CSD, current source density; EPSC, excitatory post-synaptic current; FFT, fast Fourier transformation; ICA, independent component analysis; ICMS, intracortical microstimulation; LFPs, local field potentials; LGN, lateral geniculate nucleus; MUA, multi-unit activity; NHPs, nonhuman primates; SOBI, second-order blind identification; tACS, transcranial alternating current stimulation.

schizophrenia, and Parkinson's disease [1–4]. However, it is still unknown how electrical stimulation modulates evoked LFPs across cortical layers. While there is a comprehensive understanding of how the phase and amplitude of electrical stimulation affects neural dynamics at the level of single neurons [5–7], the extension of this knowledge to cortical layers requires careful investigation. Investigating how electrical stimulation affect layer-specific LFPs is crucial for bridging the gap between individual neural activity and the synchronized response of neural populations to external currents across layers. This understanding is also important for translating insights from layer-specific LFPs to human EEG, which represents the cumulative activity of LFPs [8]. To achieve this, we recorded LFPs while applying electrical stimulation in nonhuman primates (NHPs). NHPs have emerged as significant models for studying the biophysical and physiological effects of electric stimulation [5,9–14], at various spatial scales due to the similarity of their cortical layers to those of humans [15].

The mammalian neocortex, cytologically characterized by its six distinct layers, forms an interconnected network crucial for processing cortical information [16,17]. In the sensory cortex, neurons in layer 4 mostly receive feedforward inputs from the thalamus, which are then strongly projected to layers 2/3 for further processing [18]. Subsequently, these inputs are forwarded to layers 5/6, where recurrent inputs to layer 2/3 originate [19]. Local field potentials (LFPs) have been recorded from those layers using multisite laminar probes to understand the microcircuit of cortical columns in the sensory cortex [20–22]. Especially, the primary visual cortex (V1) has been widely explored due to its well-characterized connectivity and distinct cell types in each layer [23–25]. Moreover, layer-specific LFPs can be effectively investigated by applying a visual stimulus, which serves as a sensory evocation to generate feedforward activation of the canonical cortical circuit [26–28]. LFP activity reveals that an early current source occurs in the input layer (layer 4), followed by a current sink in the superficial layers (layers 1–3) and deep layers (layers 5/6).

Sensory-evoked LFPs show distinct spatiotemporal patterns across cortical layers. One notable observation is that evoked LFPs in deeper layers are stronger than those in superficial layers [21,29]. A previous study demonstrated that flash-evoked LFPs increase with more depth in the rat brain in response to visual stimuli [30]. This pattern emerges because sensory input is primarily injected into layer 4, as well as due to cytological structure differences among the layers. For instance, large pyramidal cells in layer 5 generate strong dipoles along the dendrites, contributing to larger LFPs [8]. In addition, the variation in firing rates across layers affects LFP amplitude, as LFPs reflect the summation of synaptic activity from neural spikes in neuron populations [8]. Cortical column modeling of mouse V1 has shown that neurons in layers 2/3 have the lowest firing rates, whilst those in layers 5/6 have the highest firing rates [31]. As a result, when thalamic input generated by sensory stimuli is injected into the model, layers 5/6 exhibit larger LFPs [32].

Here, we record LFPs using laminar probes across all layers of V1 in two lightly anesthetized NHPs, employing a flash visual stimulus to evoke sensory responses while simultaneously applying 1.5 Hz alternating current (AC) transcranially to the occipital lobe. We chose transcranial alternating current stimulation (tACS) due to its

noninvasive nature and widespread use in human studies. Its ability to induce phase-dependent neural responses makes it a suitable tool for translating findings from animal to human applications [33]. We first suppress stimulation artifacts while preserving LFPs using an independent component analysis (ICA) algorithm. We demonstrate that electrical stimulation selectively increases neural activity in deeper cortical layers (layers 4–6), as evidenced by enhancements in both LFPs and multi-unit activity (MUA). Furthermore, the amplitude of LFPs is modulated in a phase-dependent manner. We observe that the LFP components, the first positive peak (P1) and negative peak (N1), show preferential increases depending on the AC phase, highlighting the phase-dependent nature of LFP modulation. We further use a cortical column model of V1 to investigate the layer-specific effects observed in vivo experiments. Our findings demonstrate how electrical stimulation modulates sensory-evoked responses in a layer-specific and phase-dependent manner, bridging the gap between previously reported single-neuron level findings and network-level cortical layer dynamics. This layer-specific understanding provides a valuable insight into the causal relationship between neural activity within cortical layers and the LFPs in response to electrical stimulation.

## Results

Capuchin monkeys ($n = 2$) were surgically implanted with a multisite probe with 23 contacts in the V1 while they sat on a primate chair, receiving flash visual stimuli generated by a light stimulator (Fig 1A and 1C). The probe was inserted perpendicular to the cortical surface, considering the alignment of the pyramidal neurons. Layers were identified by analyzing the distribution of current source density (CSD) and electric field (Fig 1B; see Materials and methods). Two sets of LFPs were recorded: one without electrical stimulation (Flash condition) and another with electrical stimulation (Flash + AC condition). The aim was to examine how electrical stimulation affects the modulation of LFPs in a layer-specific manner. We removed the AC artifacts in laminar recordings using the ICA algorithm [34] while preserving the visual evoked LFPs (Fig 1D). In the Flash + AC condition, the power spectrum density showed a complete rejection of the AC frequency (1.5 Hz) compared to the power of electrical stimulation (Fig 1F).

### Effects of electrical stimulation on LFP modulation in layer-specific manner

We recorded visual-evoked LFPs across contacts and normalized them relative to the largest amplitude of LFPs in layers 5/6. In both capuchin monkeys, LFPs have a higher amplitude in the deeper layers (layers 4–6) compared to the superficial layers (layers 1–3) (Figs 2A and A in S1 Text). The increased strength of LFP amplitudes at deeper depths may be attributed to the higher synaptic and neural activity occurring in deeper layers. These findings are in line with previous studies that have recorded LFPs in the sensory cortex [35,36]. Notably, the amplitude of LFPs after 100 ms from the onset of the visual stimulus was higher under the Flash + AC condition compared to the Flash condition, especially in the deeper layers. This difference is evident in the changes in LFP components, P1 and N1, across layers (Fig A in S1 Text). For monkey 1, the mean N1 amplitudes in layers 5/6 were $0.59 \pm 0.1$ (Flash + ES condition) and $0.47 \pm 0.12$ (Flash condition), while the mean P1 amplitudes were $0.13 \pm 0.08$ and $0.12 \pm 0.09$, respectively. A similar trend was observed in monkey 2, with the mean N1 amplitudes of $0.38 \pm 0.12$ with AC and $0.28 \pm 0.06$ without AC, and the P1 amplitudes of $0.10 \pm 0.09$ and $0.06 \pm 0.06$, respectively.

We performed a cluster-based permutation test to determine the statistical significance between Flash and Flash + AC conditions. We found a significant difference between the two conditions across contacts ranging from layer 4AB to layers 5/6 in the time range between 100 ms and 250 ms for monkey 1 ($p = 4.99 \times 10^{-4}$) and monkey 2 ($p = 9.99 \times 10^{-4}$). No significant clusters were observed during the time range of 0–100 ms (Figs 2B and B in S1 Text). Unlike monkey 1, in monkey 2 we observed a significant difference between the two conditions across contacts in layers 2/3 between 100 and 140 ms. This difference may be due to the lower volume conduction effect of LFPs from the input layer to layers 2/3 in monkey 2 during the Flash condition [37]. The results show that the amplitude of LFPs evoked by sensory stimuli increased after 100 ms, especially in the deeper layers (layers 4–6). This observation suggests that electrical stimulation enhances the amplitude of the N1 component.

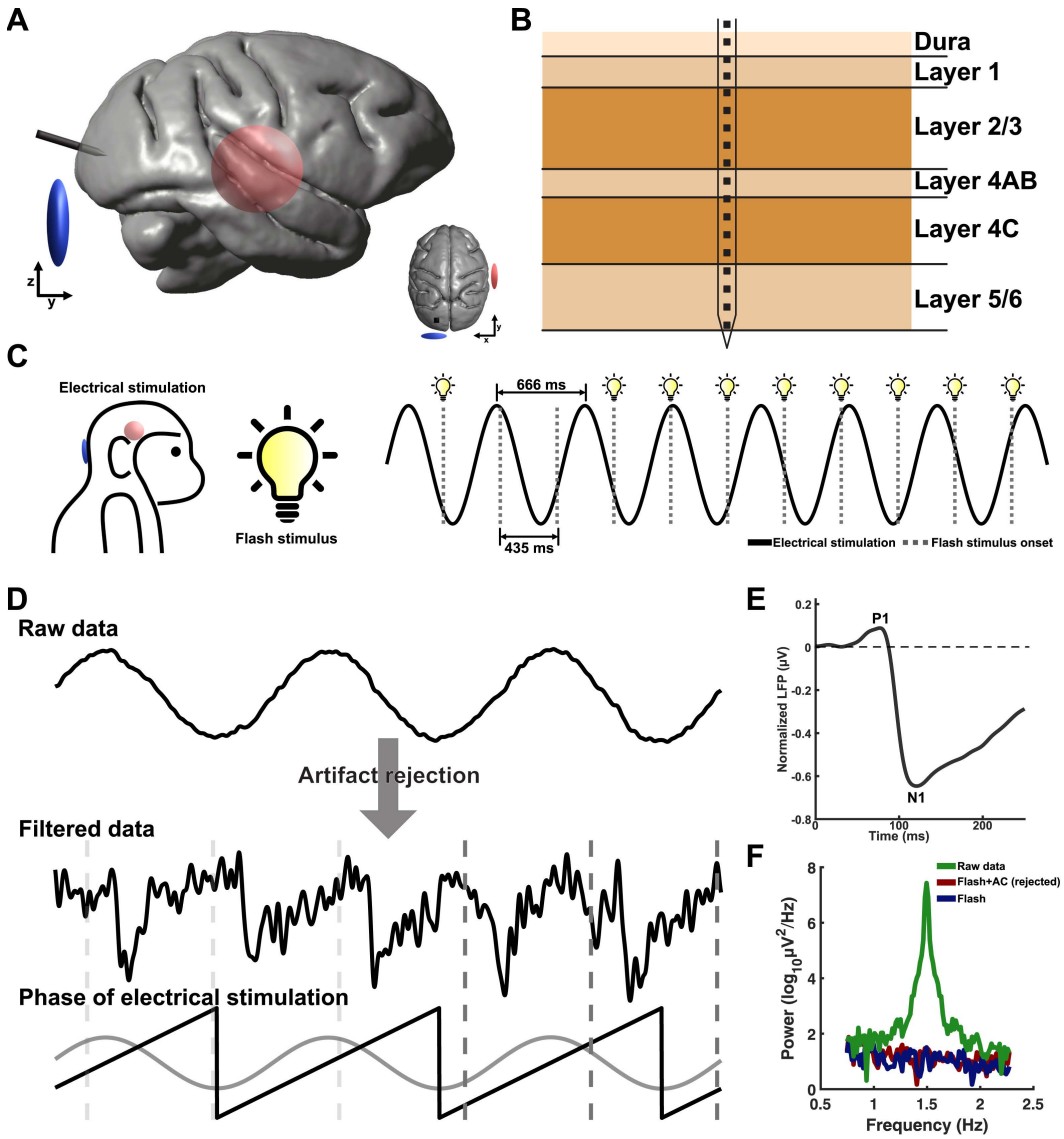

**Fig 1. Experimental design for obtaining visual-evoked LFPs in V1 layers during 1.5 Hz alternating current. A)** A multisite laminar probe was inserted into the capuchin monkey's V1. One electrical stimulation electrode (blue) was positioned on the scalp near the probe, while the other one (red) was placed over the right temporal area. **B)** Schematic illustration showing the placement of the contacts across cortical layers in V1. **C)** Visual stimuli were delivered to the monkeys at a frequency of 2.3 Hz using a high-intensity light stimulator. Concurrently, electric current oscillating at 1.5 Hz was injected via the electrodes. **D)** Raw LFPs were processed to remove AC artifacts while preserving visual-evoked LFPs. A 1.5 Hz AC signal was extracted by applying a bandpass filter between 0.5 and 2 Hz. The phase of AC was calculated using the Hilbert transformation for further phase dependency analysis. **E)** Illustration of visual-evoked LFP, with the first positive peak labeled as P1 (approximately 75 ms from the visual onset) and the first negative peak labeled as N1 (approximately 120 ms). **F)** The power spectrum density shows that the algorithm for the artifact removal effectively eliminates AC artifacts in the Flash + AC condition.

## Phase dependency of layer-specific LFPs during electrical stimulation

Next, we performed a phase dependency analysis to investigate whether the amplitude of the LFP components, P1 and N1, changes depending on the phase of AC. The trial-based LFPs were sorted into 20 phase bins based on the phase of AC, which was determined by the phase at the onset of the visual stimulus. Then, we determined the P1 and N1

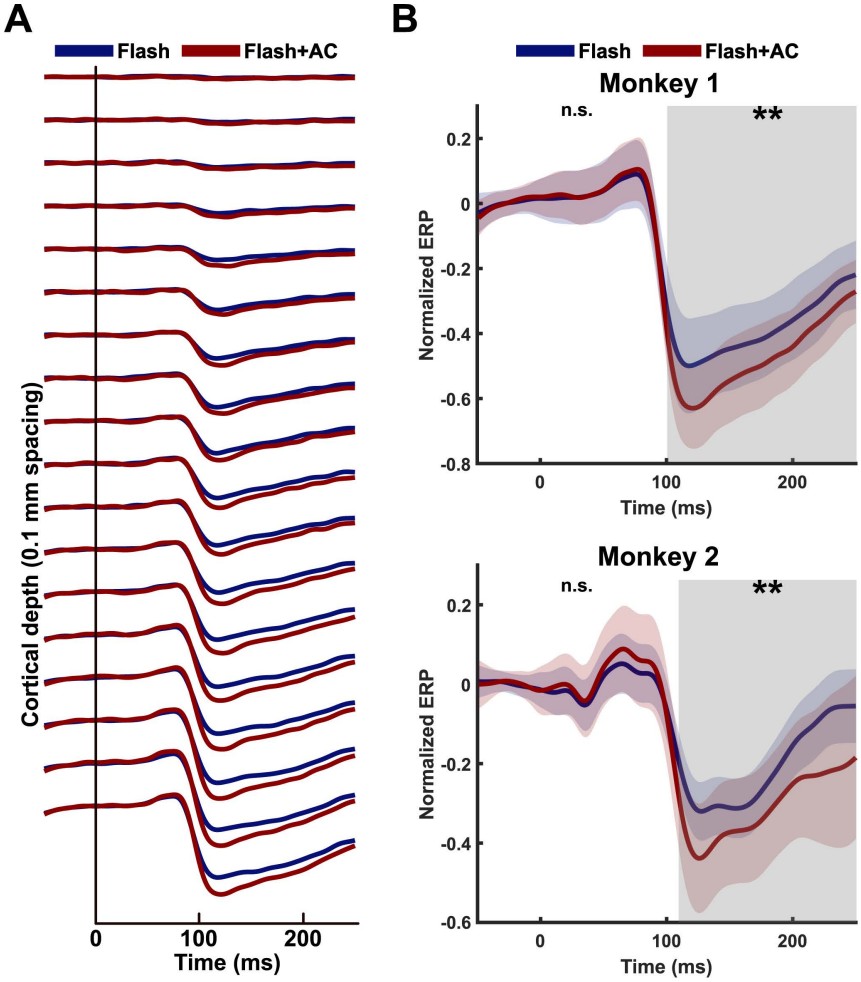

**Fig 2. Effects of electrical stimulation on layer-specific visual-evoked LFPs in V1. A)** Local field potentials (LFPs) recorded using a multisite probe for monkey 1. LFPs along contacts (0.1 mm spacing) during the Flash condition (blue line) and the Flash + AC condition (red line). LFPs were normalized relative to the last contact, which has the largest LFP, followed by averaging them across trials at each contact. Time indicates the duration from the flash visual stimulus onset. **B)** LFPs in layers 5/6 for both capuchin monkeys. Normalized LFPs were averaged across trials and contacts within layers 5/6. Thick lines and shades represent the averaged LFP and standard deviation, respectively. The cluster-based permutation test determined significant differences between the Flash and Flash + AC conditions across contacts in the time range from 0 to 250 ms. Significant differences occurred within the contacts in layers 4–6 after 100 ms from the onset of the flash visual stimulus. The gray shade represents time windows where significant differences were observed (**$p < 0.01$; n.s., not significant).

components from LFPs, followed by averaging them across trials for each layer and phase bin. Our findings show that the P1 component was increased during specific phases of AC for deeper layers in monkey 1. The phase dependence of the P1 amplitude was strongly distributed in a bimodal circular pattern in layers 4–6 (Fig 3B). The circular distribution of the P1 amplitude showed bimodal mean directions across different layers: −145° for layer 1, −135° for layers 2/3, −114° for layer 4AB, −118° for layer 4C, and −113° for layers 5/6. The bimodality in the N1 amplitude was less pronounced than in P1. The mean directions of the bimodal distribution of the N1 amplitude were as follows: −40° for layer 1, −61° for layers 2/3, −41° for layer 4AB, −37° for layer 4C, and −40° for layers 5/6. Monkey 2 exhibited a unimodal circular distribution of the LFP components based on AC phases, with a similar trend as in monkey 1, showing a strong directionality for the P1 amplitude (Fig 3C). The unimodal mean direction of the P1 and N1 amplitudes was −159° and

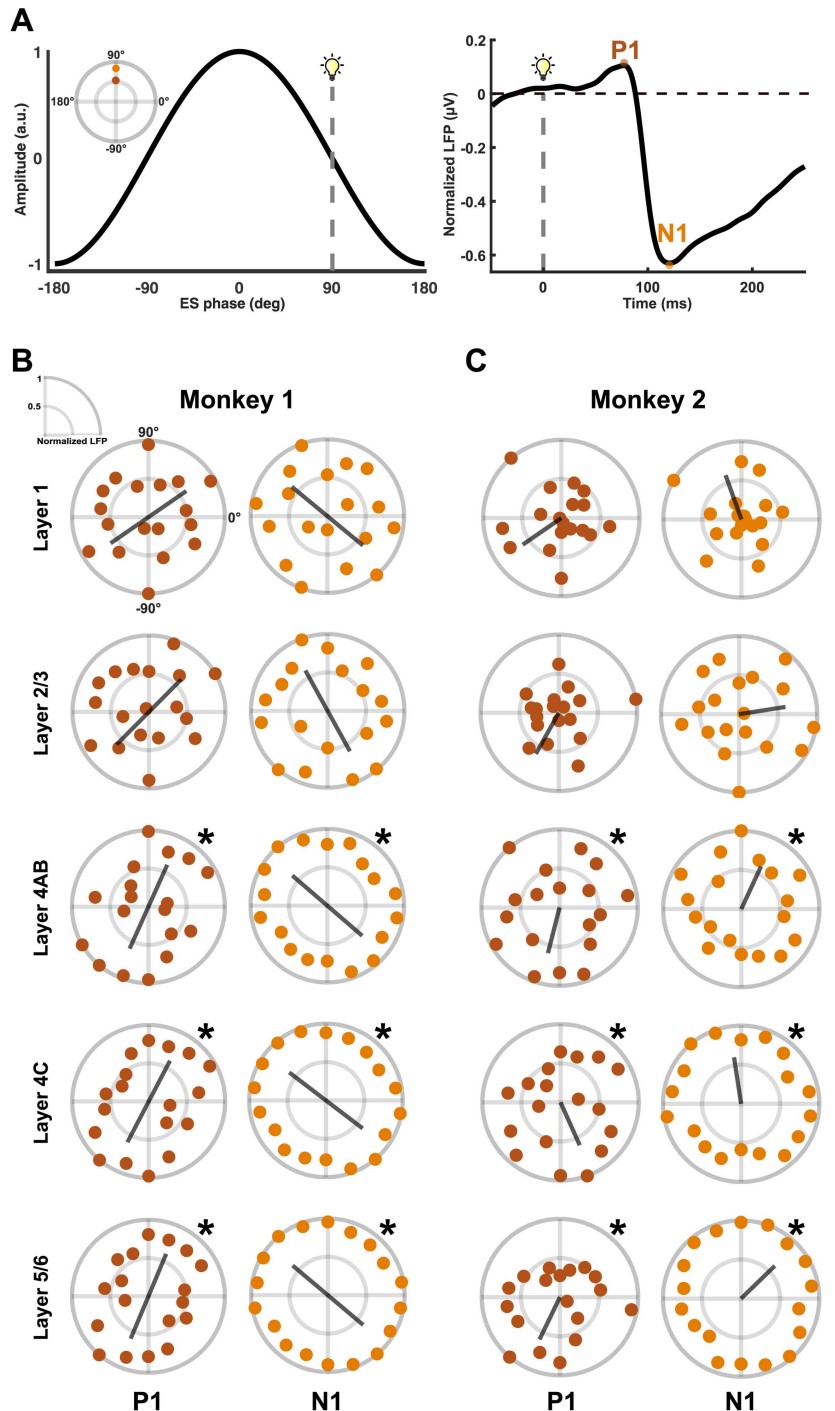

**Fig 3. Layer-specific phase dependency of LFP components during electrical stimulation.** Phase dependency of the amplitude of LFP components, P1 and N1, to external stimulation. The P1 and N1 components were sorted into 20 phase bins, followed by taking trial- and phase bin-averages for each layer. The gray lines represent the mean direction of the phase preference of LFP amplitudes. **A)** Illustration of the sorting of LFP components based on the AC phase. For instance, in a single trial, the visual stimulus was presented at a time corresponding to an AC phase of 90° (left). For each trial, the amplitudes of the P1 and N1 components were calculated and sorted into the phase bin that includes 90° (right). **B)** Circular distributions of P1 and N1 amplitudes according to the AC phase show a bimodal characteristic for monkey 1. **C)** Circular distributions show a unimodal characteristic for monkey 2. A permutation test revealed significant directional preferences in P1 and N1 amplitudes with respect to the phase of AC only within deeper layers (layers 4–6) (*$p < 0.05$) for both capuchin monkeys.

100° for layer 1, −86° and 30° for layers 2/3, −94° and 74° for layer 4AB, −59° and 107° for layer 4C, and −110° and 53° for layers 5/6.

We performed a permutation test for each layer to investigate the significance of the amplitude changes in LFP components relative to the phase of AC. Specifically, we compared the vector length in the mean direction of the original data to the vector lengths obtained from the permutation procedure. The vector length quantifies the strength of phase-locking across AC phases [38]. The P1 and N1 amplitudes show a preferred directionality with respect to the phase of AC only in deeper layers (layers 4–6). No significant effects were observed in superficial layers (layers 1–3) for both capuchin monkeys (Fig C in S1 Text). The amplitude of P1 in deeper layers was larger during the phase, in which the amplitude of N1 was smaller for both capuchin monkeys. This raises the question of whether the observed phase dependency is caused by electrical stimulation or an inherent physiological phenomenon. For instance, it is possible that the onset of the visual stimulus happens to coincide with a specific bio-signal oscillating at 1.5 Hz. As a control analysis, we performed the same phase dependency calculation on the data from the Flash condition using a virtual AC at 1.5 Hz. We found no significant phase preference of the amplitude of LFP in both P1 and N1 components was found across all layers (Fig D in S1 Text). These results suggest that the amplitude of the LFP component is selectively modulated in a layer-specific manner depending on the phase of an external current.

## Biophysics of electrical stimulation across cortical layers

To investigate the relationship between LFP changes and the biophysics of electrical stimulation, we measured the electrical voltage and electric field across cortical layers (see Table A in S1 Text). In monkey 1, the electric field exhibited an initial increase from layer 1, reaching a considerable peak in layers 2/3, and thereafter decreased in deeper layers (Fig 4A). The average electric fields were as follows: 0.62 V/m in layer 1, 2.53 V/m in layers 2/3, 1.18 V/m in layer 4AB, 0.62 V/m in layer 4C, and 0.37 V/m in layers 5/6. Monkey 2 showed a similar electric field distribution across cortical layers. However, in contrast to monkey 1, higher electric fields were delivered to deeper layers (Fig 4B). The average electric field was to be 0.97 V/m in layer 1, 2.76 V/m in layers 2/3, 1.6 V/m in layer 4AB, 1.59 V/m in layer 4C, and 1.42 V/m in layers 5/6. Interestingly, we did not observe the effects of electrical stimulation on the LFP modulation in layers 2/3, despite their relatively high electric field. This finding suggests that neurophysiological properties of cortical layers have a stronger effect on LFP modulation than the biophysical properties of electric stimulation.

## Multi-unit activity across cortical layers

We additionally conducted MUA analysis in monkey 2 to assess how neural population activity is modulated by AC. Fig 5A shows the PSTHs across layers under the Flash and Flash + AC conditions, demonstrating that neural activity was higher in deeper layers compared to the superficial layers in both conditions. Notably, AC led to a comparable increase in firing rates in layers 5/6, while the Flash condition showed relatively stronger activation in layer 4AB. Statistical analysis confirmed that MUA firing rates in layers 5/6 were significantly higher than those in layers 2/3 under both the Flash ($6.83 \pm 5.35$ versus $5.32 \pm 2.22$ spikes/s, $p = 2.03 \times 10^{-6}$) and Flash + AC ($9.85 \pm 6.09$ versus $5.03 \pm 2.03$ spikes/s, $p = 6.09 \times 10^{-42}$) conditions (Fig 5B). Furthermore, when comparing across conditions, firing rates in layers 5/6 were significantly higher in the Flash + AC condition than in the Flash condition ($p = 2.54 \times 10^{-12}$), whereas no significant difference was observed in layers 2/3. Fig 5C shows the PSTHs across layers between the peak and trough phases of AC. Although firing rates tended to be slightly higher during the trough phase compared to the peak phase, no significant differences were found ($p = 0.36$ for layers 2/3; $p = 0.51$ for layers 5/6; see Fig 5D). Firing rates were $4.97 \pm 1.92$ spikes/s (peak) and $5.24 \pm 2.08$ spikes/s (trough) in layers 2/3, and $9.42 \pm 6.25$ spikes/s (peak) and $10.01 \pm 6.17$ spikes/s (trough) in layers 5/6.

 

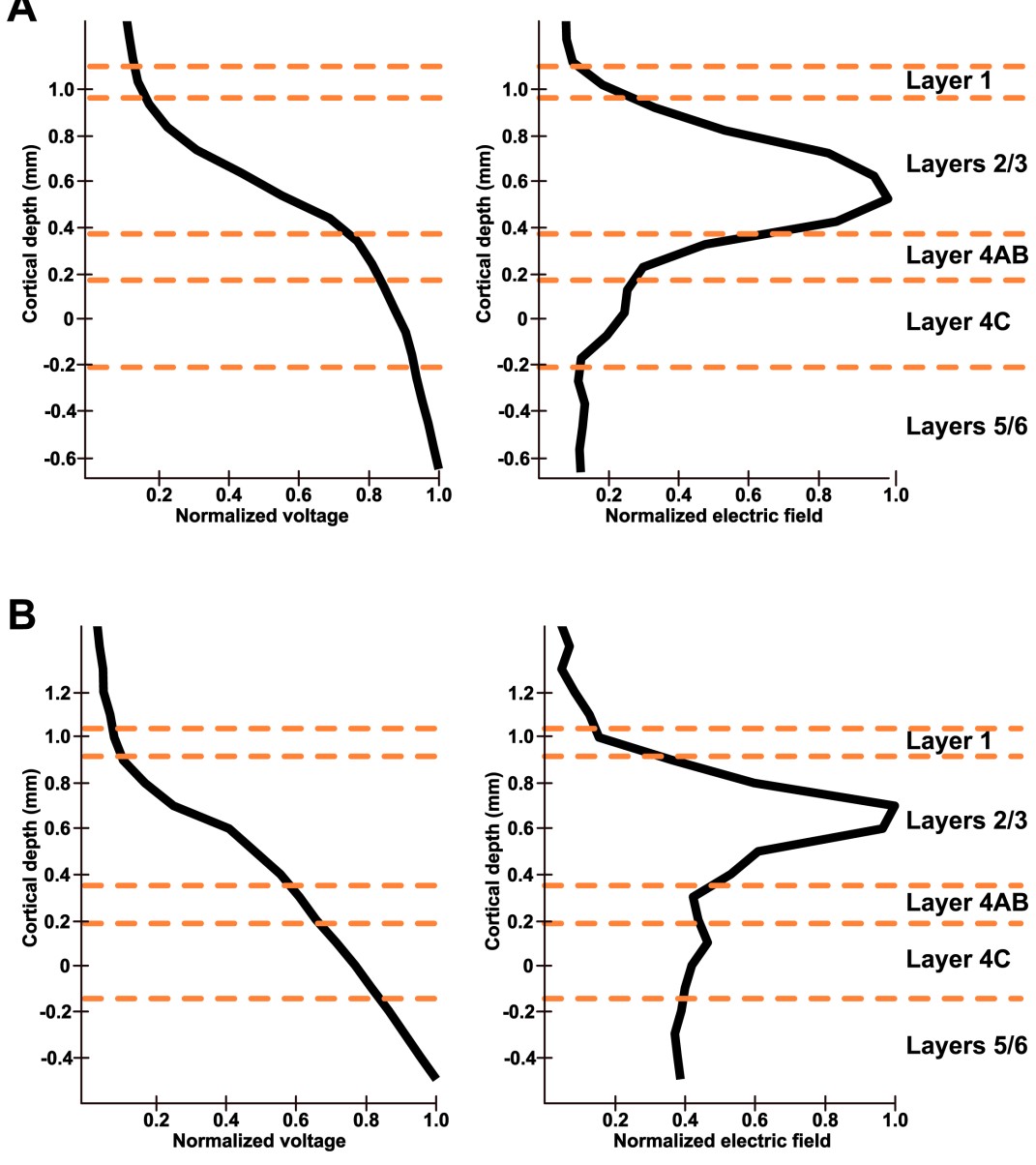

**Fig 4. Biophysics of electrical stimulation in V1 layers.** Distributions of electrical voltage and electric field across cortical layers in **A)** monkey 1 and **B)** monkey 2. The electrical voltage and electric field values were normalized to their maximum values. The electric field was calculated by taking the gradient of the voltage along the direction of the laminar probe. For both capuchin monkeys, the electric field began to increase from layer 1, reaching the maximum in layers 2/3, and then decreased in deeper layers.

### Neural activity in cortical column model

In order to further elucidate how observed layer-specific changes in LFPs can be explained, we extended a cortical column model of V1 [31] to integrate AC stimulation. Before applying AC, we investigated how neural activity arising from the flash stimulus varies across layers. Our model shows that excitatory neurons in the deeper layers have a distinct firing response around 60 ms after the visual stimulus onset (Fig 6A). Firing rates are higher in layers 5/6, followed by layers 4 and 2/3 (Fig 6B). This trend is in line with stronger LFPs in layers 5/6 (Fig F in S1 Text). Firing rates in layers 5/6 were

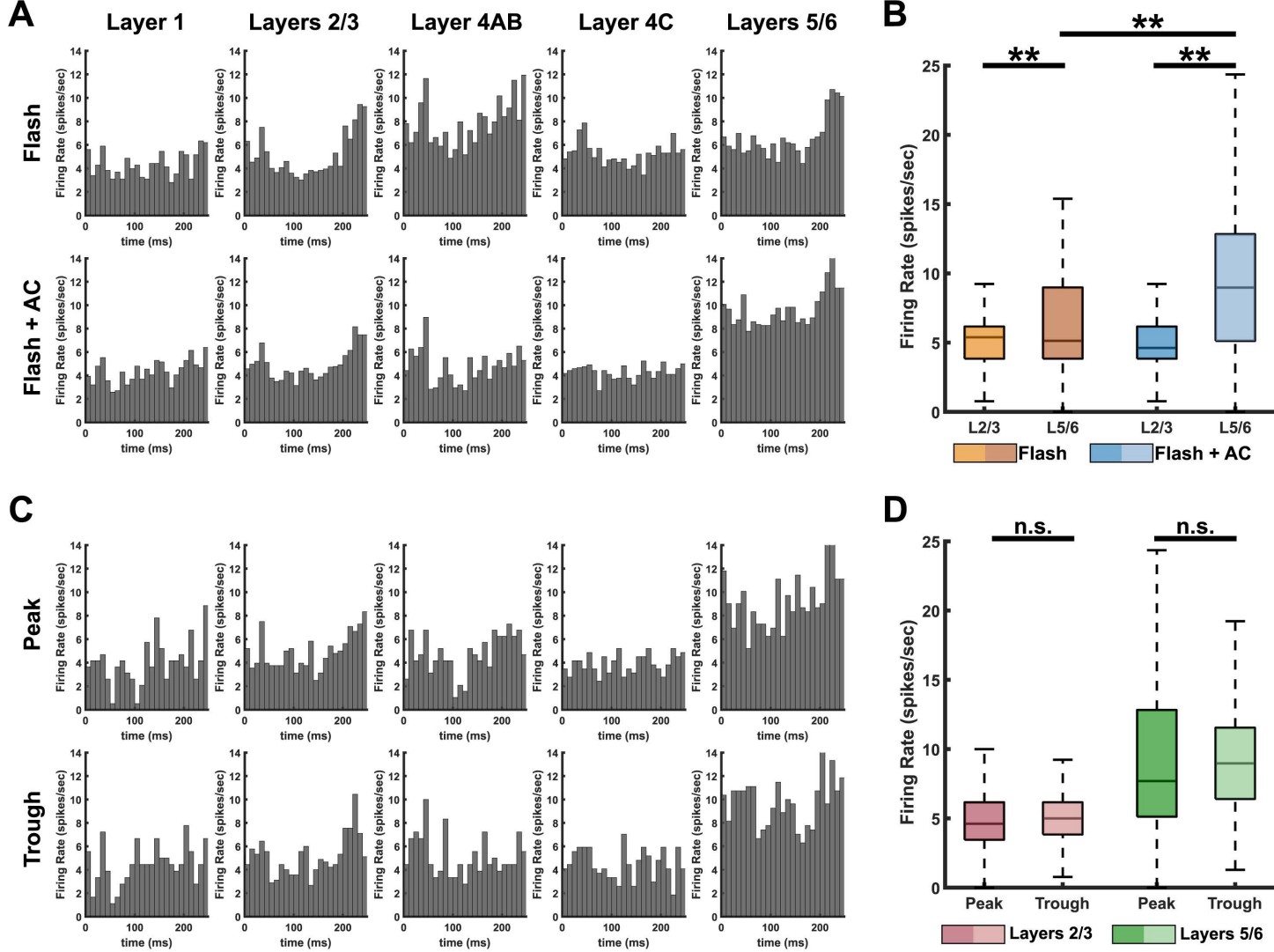

**Fig 5. Multi-unit activity responses across cortical layers in monkey 2. A)** Peri-stimulus time histograms (PSTHs) across layers under Flash and Flash + AC conditions, aligned to visual stimulus onset (0 ms) and extending to 250 ms. Each bar represents the firing rate (spikes/s) within a 10 ms time bin. **B)** Comparison of time-averaged firing rates between superficial layers (layers 2/3) and deeper layers (layers 5/6) under both conditions. Firing rates of MUA were significantly higher in layers 5/6 for both conditions (paired $t$ test, $p < 0.01$), and AC significantly increased firing rates only in layers 5/6 (unpaired $t$ test, $p < 0.01$). **C)** PSTHs across layers for trials aligned to the peak and trough phases of AC. **D)** Comparison of time-averaged firing rates between superficial layers and deeper layers across AC phases. No significant differences were found. Underlying data for this figure are provided in S1 Data.

determined by averaging the rates from both layers to align with in vivo findings, whereas layer 1 was excluded due to the absence of excitatory neurons.

Next, we applied AC in the model to investigate the layer-specific phase dependency observed in in vivo LFPs. Interestingly, our model showed that the P1 (about 50 ms) occurred before neural firing (Figs 6A and F in S1 Text), suggesting that factors other than neural firing may contribute to the strong phase preference of P1 in our observations. We hypothesized that the membrane current ($I_{mem}$) induced by thalamic input around 50 ms (Fig F in S1 Text) in the basal dendrites, mostly located in layers 5/6, is modulated in a phase-dependent manner. Given that $I_{mem}$ is influenced by the membrane

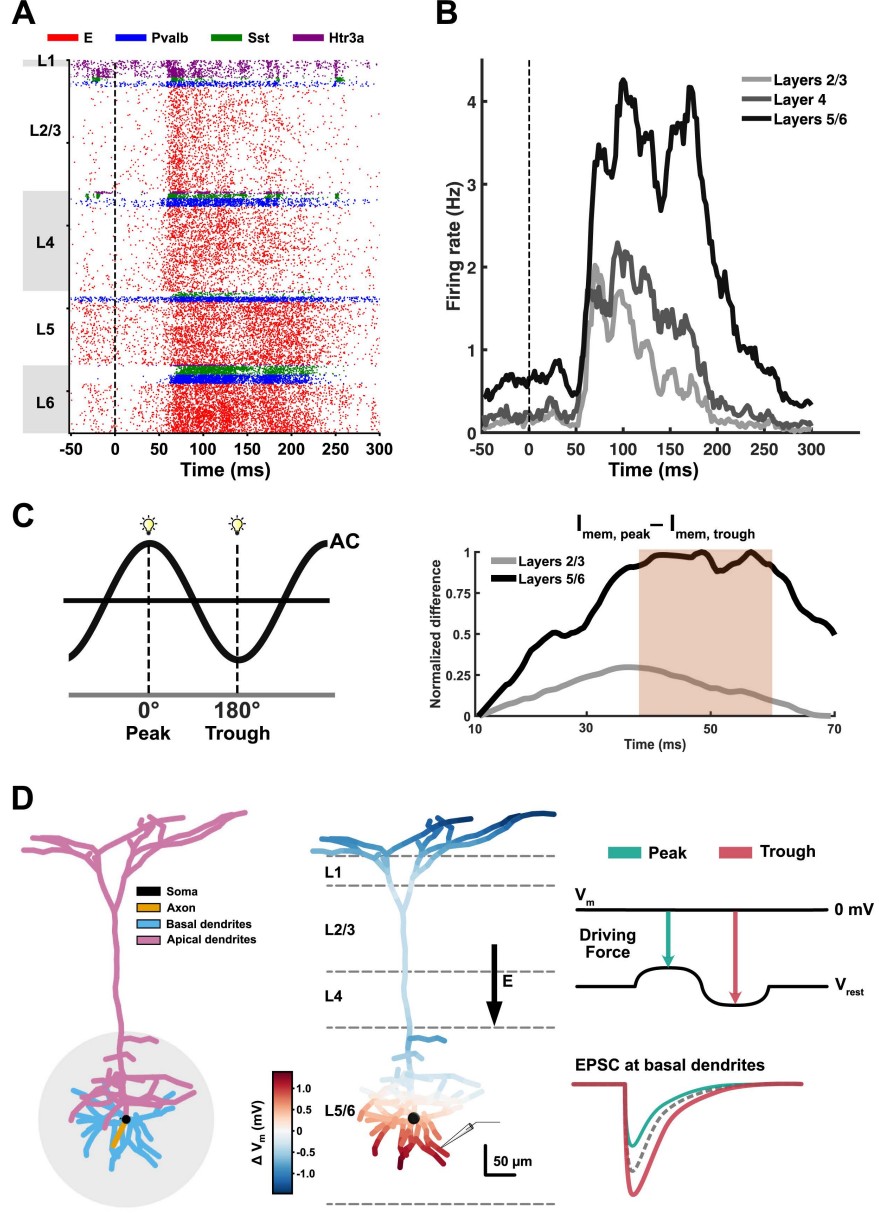

**Fig 6. Neural activity in the cortical column model of V1 during flash stimuli. A)** Raster plot showing the neural spiking of different types of neurons across layers. The red dots represent excitatory neurons, while the others represent parvalbumin-positive interneurons (blue), somatostatin-positive interneurons (green), and 5-HT3a receptor-positive interneurons (purple), respectively. **B)** Averaged firing rate of excitatory neurons across layers over time, calculated using a 2 ms time bin. Dash lines indicate the onset of flash stimuli. **C)** Membrane current from the basal dendrites of 500 neurons in both layers 2/3 and layers 5/6 was calculated under two conditions: when a flash stimulus was applied at the peak phase and the trough phase of AC (left). The difference between the membrane currents in the peak condition ($I_{mem, peak}$) and trough condition ($I_{mem, trough}$) is the highest during the period of P1 in LFP around 50 ms. The red shade represents the period of P1. **D)** Schematic illustration explaining the phase dependency. The synaptic input from the lateral geniculate nucleus (LGN) enters the region adjacent to basal dendrites in deeper layers (gray circle), which are highly responsive to this input (left). When the peak phase of AC flows in a downward direction (middle), the membrane potential in basal dendrites is depolarized, leading to a weaker driving force. It results in weaker (less negative) excitatory postsynaptic current (EPSC) affecting change in LFPs and vice versa during the trough phase.

potential, which is responsive to the phase of the AC, or the polarity of the electric field, these phase-dependent changes in $I_{mem}$ likely contribute to the phase dependency of LFP. To test this, we calculated $I_{mem}$, an indirect feature determining the LFP deflection, when a flash stimulus was applied at either the peak or trough phase of AC. Our results show that the difference in $I_{mem}$ between peak and trough conditions was positive during the P1 period, with a stronger difference in the deeper layers (Fig 6C). These findings suggest that the peak phase of AC produces a more positive (or less negative) $I_{mem}$, leading to a more positive LFP relative to the trough condition. Furthermore, they elucidate the distinct phase preference of P1 specifically in the deeper layers rather than the superficial layers, as shown in Fig 3B and 3C. On the other hand, the trough phase of AC induces a more negative $I_{mem}$, leading to a larger negative LFP compared to the peak condition. The membrane current during the peak phase consistently exceeds that in the trough phase, resulting in opposite phase preferences for the positive and negative LFP deflections, as observed in our in vivo recordings.

## Discussion

In this study, we demonstrate that visually evoked LFPs are modulated by AC in a layer-specific and phase-dependent manner, with stronger modulation observed in deeper cortical layers. Our cortical column model further supports this finding by showing that neurons in deeper layers experience larger phase-dependent changes in driving force, which helps explain in vivo observations. These findings provide direct evidence for how electric fields selectively influence cortical microcircuits.

Electric current stimulation can alter the membrane potential of neurons, causing either depolarization or hyperpolarization [7,39,40]. LFPs reflect extracellular signals generated by transmembrane currents, primarily arising from postsynaptic potentials from synchronized neuronal populations [8,41]. Consequently, external electrical stimulation that modifies membrane polarization also influences LFPs. Our results show that visual-evoked LFPs during AC are enhanced compared to no stimulation, but this effect is only observed in deeper layers. These findings suggest that neurons in deeper layers are more responsive to AC than those in superficial layers, likely due to their distinct anatomical and physiological properties. Layer 4 is characterized by its high neuronal density [32,42], and layer 5 includes larger pyramidal cells compared to layers 2/3 [42,43]. These structural differences may contribute to a stronger response to AC due to the increased number of neurons producing synchronized activity. Indeed, deeper layers exhibit higher firing rates in response to visual stimuli compared to layers 2/3 [44], and neuronal synchronization is generally lower in superficial layers [45,46]. Consistently, our MUA recordings showed that firing rates in layers 5/6 were significantly higher than those in layers 2/3 (Fig 5). This effect is further supported by our cortical column model, showing that the neural firing rate in deeper layers is higher than in layers 2/3 (Fig 6B).

In white matter, LFPs did not show any modulation of evoked potentials by electrical stimulation (Fig E in S1 Text). A cluster-based permutation test confirmed that there was no significant difference in LFPs between the Flash and Flash + AC conditions. Similarly, phase dependency analysis indicated that the amplitudes of P1 and N1 did not depict any preference based on the phase of AC (Fig E in S1 Text). These findings suggest that electrical stimulation selectively modulates LFP activity in the deep layers of gray matter. White matter consists mainly of complex fiber tracts, which lack the sufficient neural activity needed to generate a response to electrical stimulation [47,48]. This highlights the anatomical and functional specificity of different cortical layers in their responsiveness to electrical stimulation.

Interestingly, the physiological effects observed in our study cannot be fully explained by the biophysics of electrical stimulation. The modulatory effects of AC stimulation are primarily driven by the electric field [39], yet we found that the electric field strength in layers 2/3 was at least twice as strong as in layers 5/6 in both capuchin monkeys (Fig 4). This suggests that certain anatomical features in layers 2/3 reduce electrical conductivity, leading to higher electric fields. Lower conductivity increases the tissue's resistance to current flow, amplifying the local electrical field. One possible explanation is the high density of blood vessels and astrocytes [49], as endothelial cells in blood vessels form gap junctions that increase electrical resistance [50]. In fact, computational simulations have demonstrated that electric fields are markedly

higher when the blood vessels are included in models during electrical stimulation [51]. Despite the stronger fields, layers 2/3 did not show modulation or phase dependency in LFPs, unlike deeper layers. Therefore, we conclude that LFP modulation is more sensitive to layer-specific neural features than to the amplitude of the electric field.

A phase preference for the P1 component only emerges in deeper layers, suggesting that neurons in layers 5/6 have a stronger neural response to specific AC phases. One possible explanation lies in the modulation of the electrochemical driving force by AC phase-induced changes in membrane potential. These alternations in the driving force affect the postsynaptic currents [52]. As the excitatory postsynaptic current (EPSC) changes, the resulting fluctuations in membrane currents lead to changes in the extracellular potential, as reflected in LFPs (Fig 6D). We focused on basal dendrites because they are the primary sites for synaptic inputs and neural integration [53]. During the peak phase of AC, basal dendrites become depolarized, leading to a weaker driving force. Conversely, during the trough condition, they become hyperpolarized, leading to a stronger driving force. This difference is reflected in the membrane currents, which are higher under the peak condition (Fig 6C). As a result, a weaker driving force produces a less negative EPSC, leading to a more positive (less negative) LFP deflection. In our in vivo experiments, the P1 component had a preferred phase around −110° for both capuchin monkeys, corresponding to the rising phase of AC. During this phase, the driving force consistently increased throughout the period of the evoked LFP following the visual flash. This effect appears to be layer-specific, as the basal dendrites of large pyramidal neurons in the deeper layer show a high response to AC, which may spread to adjacent layer 4 due to volume conduction [54]. However, MUA recordings revealed a significant increase in firing rates in layers 5/6 during AC, suggesting that the observed LFP modulation likely reflects primarily genuine neuronal activation rather than passive electrical spread.

Our explanation provides insight into why the P1 and N1 components have distinct preferred phases. Importantly, the temporal relationship between LFP components and spikes supports their functional dissociation. The P1 peak appears prior to the onset of pyramidal neuronal firing, suggesting that it is more likely driven by EPSCs rather than by spiking activities (Fig F in S1 Text). Unlike P1, the N1 peak appears to be more associated with neuronal firing, as it occurs after the onset of evoked firing of pyramidal neurons in V1. Indeed, it shows a strong correlation with the firing rate as depth increases (Fig F in S1 Text). Both LFPs and spikes are shaped by synchronized excitatory synaptic inputs, especially in deeper layers where our recordings were made. LFPs primarily reflect postsynaptic currents in the basal dendrites, which are spatially close to the soma and directly influence spike generation. Previous studies have shown that excitatory inputs to these basal dendrites contribute strongly to both the LFP and the initiation of spiking in layer 5 pyramidal neurons [55]. Therefore, a strong excitatory driving force not only generates a larger LFP deflection (N1) but also increases the likelihood of neuronal firing. Together, neurons in V1 become less excitable during the preferred phase when the P1 peak is stronger, likely due to a weak driving force. As a result, fewer neurons fire, leading to a reduced N1 amplitude. On the contrary, when the P1 peak is lower, neurons are more excitable, increasing the likelihood of firing and producing a larger N1 amplitude.

Even though both LFPs and our cortical column model demonstrated phase-dependent changes in neuronal activity, MUA firing rates did not show clear phase dependency. This dissociation presumably reflects the different sensitivities of these two measures. LFPs primarily capture subthreshold synaptic currents and membrane potential fluctuations, which are highly susceptible to AC-induced oscillations, whereas MUA represents spiking activity resulting from suprathreshold depolarization. While AC effectively modulated the strength of synaptic currents, several factors may have limited its ability to induce phase-dependent modulation of MUAs, such as the relatively small amplitude of membrane oscillations caused by AC, strong sensory-driven firing triggered by visual stimuli, and intrinsic variability within the cortical network. As a result, phase-dependent modulation of LFPs arises from entrained subthreshold synaptic activity, while spiking activity, as reflected in MUA, may remain insensitive to AC phases under the current experimental conditions.

There are differences in the results between two animals. In monkey 1, the average P1 component remains unchanged with and without AC, with certain phase bins showing a decreased P1 amplitude, contributing to a bimodal pattern (Fig I

in S1 Text). Conversely, monkey 2 showed an increased P1 component across most phases. We suggest that variation is due to differences in electric field strength, as the field in the deeper layers of monkey 2 was more than twice as strong as that in monkey 1 (Table A in S1 Text). With a weaker electric field, sensory evoked responses remained influenced by the phase of spontaneous low-frequency oscillations [56,57], as the external stimulation was insufficient to override the intrinsic phase-dependent excitability, resulting in two competing preferred phases: one phase-locked to spontaneous activity and the other driven by the externally applied AC. While our model provides a generalizable framework showing how tACS phase modulates neural activity and accounts for the majority of the phase-dependent effects observed in the monkeys, it does not fully capture the bimodality seen in monkey 1. This is likely because the model does not incorporate spontaneous activity or microcircuit-level variability. Note that our goal was to model the general interaction between sensory-evoked neural activity and the phase-specific effects of AC. Nevertheless, we suggest that extending the model to include spontaneous network dynamics could better address inter-variability and represents an important direction for future research.

Our findings suggest that future brain stimulation applications could benefit from approaches to selectively target deeper cortical layers to enhance modulation effects. Intracortical microstimulation (ICMS) enables selective modulation of deeper cortical layers [58,59], which are more responsive to electrical stimulation than superficial layers [60–62]. Moreover, previous studies have shown that the phase of ongoing brain rhythms critically influences the effects of ICMS on neural activity [63,64], with phase-specific stimulation shown to modulate LFPs and affect behavioral outcomes in NHPs [65]. ICMS has also been shown to elicit sensory perceptions in humans [58]. Together, these findings suggest that phase-dependent stimulation targeting deeper layers may more effectively facilitate behavioral outcomes in humans. On the other hand, tACS provides non-invasive, subthreshold stimulation and is widely employed in human applications, despite its lack of spatial focality. To overcome this limitation, multi-channel transcranial electrical stimulation would be a potential approach that allows for more precise and focal targeting of specific regions [66]. Another promising technique is temporal interference stimulation, which has been shown in both in silico and in vivo studies to selectively stimulate deeper brain regions [67–70]. While these approaches hold great potential, further studies are needed to better understand their effectiveness in targeting deeper layers.

In our experiments, the capuchin monkeys were lightly anesthetized to minimize interference from other neural activities. While stimulation in the awake conditions may yield different effects, sensory-evoked LFPs during anesthesia are well-preserved compared to the awake state [71]. To improve the separability of AC stimulation from ongoing LFPs, we used low-frequency stimulation to minimize the potential influence of LFP modulation caused by the entrainment of intrinsic oscillations, such as alpha, beta, and gamma rhythms [72,73]. While rhythmic visual stimuli may induce neural entrainment, such effects would be consistent across conditions. Accordingly, baseline correction would minimize the influence of visual-induced entrainment. This allows us to more precisely isolate the LFP modulation induced by AC, which is the main focus of our study. In addition, the choice of 1.5 Hz AC and 2.3 Hz flash stimuli effectively jittered the phase, ensuring nearly equal numbers of trials across phase bins (Fig G in S1 Text). Nevertheless, it would be an intriguing topic to apply different frequencies of AC to investigate whether phase-dependent LFP modulation arises in a frequency-specific manner. In our model, we used a well-established cortical column model from the rat. While anatomical differences in cortical layers such as sublaminar distinctions and differences in layer thickness exist between rats and monkeys [74], fundamental principles of laminar processing, including balanced excitatory and inhibitory, canonical computations, and early visual processing pathways evoked by the flash stimulus used in our study, are conserved across mammalian species [17,75]. These shared features support our comparative approach and the relevance of using the rat model to help interpret our monkey data. While our approach relies on CSD profiles and anatomical references to define cortical boundaries, alternative methods, such as those based on spike activity [76] or other electrophysiological measures [77], could provide insights into layer-specific distinctions and their relationship to LFP modulation. This would be an interesting direction for future studies. Our findings rely on the successful rejection of tACS artifacts in the LFPs. Although residual

artifacts could have an influence, we consider the possibility unlikely due to the consistent outcomes across cortical layers and the absence of effects in white matter. Additional recording modalities, such as single-unit activity, could further strengthen our findings. Additionally, extending cortical column models from small animals to humans would be essential for translating our findings and enhancing our understanding of human applications. We also observed some differences in LFP modulation between the two monkeys. This variance might originate from differences in the electric field strength across the layers. In monkey 2, the electric field in deeper layers was somewhat higher than in monkey 1, resulting in a more pronounced increase in LFP and displaying a strong unidirectional phase preference in LFP modulation. Our group has previously demonstrated dose-dependent neural entrainment during tACS in resting-state NHPs [6]. Future studies could benefit from investigating dose-response relationships to further clarify the link between current intensity and LFP modulation.

In conclusion, our findings show that sensory-evoked LFPs are modulated by electrical stimulation in a layer-specific and phase-dependent manner. This modulation was predominantly observed in deeper layers, where large pyramidal neurons are more responsive to external electric fields. Our cortical column model suggests that phase-dependent changes in the driving force within these layers can explain the observed effects. This study advances our understanding of how electric fields interact with cortical microcircuits and highlights the influence of layer-specific properties on LFP modulation. These insights can inform future strategies to optimize stimulation parameters, targeting specific layers to improve the therapeutic efficacy of neuromodulation.

## Materials and methods

### Subjects

Two female capuchin monkeys (*Cebus apella*), each aged 15 years and weighing between 1.5 and 3 kg, were included in the current study. All procedures were approved by the Institutional Animal Care and Use Committee of the Nathan Kline Institute for Psychiatric Research (IACUC protocol number: AP2014-510). Surgical procedures corresponded to the methods that were previously established [78]. Under general anesthesia, a craniotomy was performed, and each monkey was implanted with a headpost and a Cilux recording chamber (Crist Instruments) positioned over the occipital cortex to aim electrode penetrations perpendicular to the primary visual cortex (Area V1). Both implants were fastened using MRI-compatible ceramic screws (Thomas Recording, GmbH). For each recording session, we lowered a 23-channel linear array electrode (U-Probe, Plexon), previously used in other studies [35,79,80], through a grid to ensure a perpendicular insertion into area V1. The electrode intercontact spacing was 0.1 mm, and the impedance range for the contacts was 300–750 kΩ. We lowered the probe until the contacts would bracket all cortical layers, as well as the area outside the cortex. We confirmed that the probe covered all cortical layers using CSD analysis, as described later in the Materials and methods section. LFPs recorded outside the cortex (i.e., dura mater) were used to establish the border of layer 1 during electrical stimulation. We verified that the electric field did not dramatically change within the area outside the cortex but began to increase noticeably upon reaching layer 1.

### Visual stimuli

We used flash visual stimuli to induce sensory-evoked LFPs across cortical layers. The stimuli were presented using a Grass strobe light photo stimulator while the monkey was seated in a primate chair, with its head fixed inside an electrically shielded chamber under dim lighting conditions. During the experiment, the monkeys were lightly anesthetized with 1%–1.5% isoflurane. The distance between the monitor and the monkey's eyes was 86 cm. Each session consisted of 200–400 visual stimuli, presented at a frequency of 2.3 Hz (one stimulus every 435 ms). The monkeys underwent two experimental sessions: first, a session without electrical stimulation (Flash condition), and then a session with electrical stimulation (Flash + AC condition). To mitigate the visual fatigue, we ensured ample rest between the two sessions.

## Laminar recording

Two capuchin monkeys underwent laminar recordings, where a multisite probe was implanted in the V1 region for both the Flash and Flash + AC conditions. Electrical signals were recorded from 23 contacts along the probe, with an inter-contact spacing of 0.1 mm. The impedance ranged from 0.3 to 0.5 megohms. We placed the reference on the tissue layer covering the dura. The signals were then amplified by a factor of 10 and filtered (bandpass DC of 10 kHz) using the preamplifier from Plexon. To obtain LFP activities, the signals were analog filtered within the range of 0.1–500 Hz, followed by downsampling to 2 kHz.

## Electrical stimulation

Electrical stimulation was delivered transcranially using a Starstim system (Neuroelectrics, Barcelona, Spain) through Ag/AgCl electrodes attached to the scalp. One electrode was placed over the occipital lobe, right under the location where laminar probes were inserted. Another electrode was attached to the right temporal area (Fig 1A). The electrodes were round, with a radius of 10 mm. The electrode montage was chosen to ensure that the electric field is more likely to be directed parallel to the laminar probes. We preliminarily calculated the maximum electric field during the experiment and found that the maximum electric field in monkey 2 was lower than in monkey 1 when the same current was applied. Thus, we decided to inject the intensity of the current (peak-to-zero) of 0.1 mA for monkey 1 and a higher current of 0.2 mA for monkey 2. The stimulation frequency was set to 1.5 Hz to allow for the generation of various AC phase values corresponding to the onset of the visual stimulus presented at a rate of 2.3 Hz (Fig 1C). Electrical stimulation was continuously applied, with a 30-second ramp-up at the beginning, maintained until the end of the last trial, and followed by a 30-s ramp-down.

## Data preprocessing

The data were analyzed using MATLAB 2022b (MathWorks). LFPs were obtained by applying a 4th-order Butterworth digital bandpass filter between 0.5 and 100 Hz to the analog-filtered signals [81], followed by downsampling to 1 kHz using the Fieldtrip toolbox [82]. We chose to apply a cut-off frequency of 100 Hz to remove possible contamination from multiunit spiking responses in the higher frequency range, up to 300 Hz [83]. Additionally, the notch filter was employed to eliminate the powerline noise at 60 Hz. LFPs were recorded using a monopolar reference placed on the dura. While alternative referencing methods (e.g., bipolar, common average, Laplacian) can enhance certain aspects of signal analysis, we specifically selected the monopolar reference to minimize phase distortion in the LFPs [84], which is critical for our phase-dependent analysis. LFPs were segmented into 300 ms epochs ranging from −50 ms to 250 ms relative to the onset of the flash visual stimulus. Then, baseline correction was performed for each epoch using the time window from −50 ms to 0 ms. For the condition with electrical stimulation (Flash + AC condition), AC artifact-filtered data, as described in the following section, was used for epoch segmentation. In each trial, the most dominant positive peak that appeared about 75 ms after the onset of the visual stimulus was identified as P1 within the time range of 55–90 ms, while the negative peak occurring around 120 ms, within the time window of 100–140 ms, was labeled as N1 across all laminar probes (Fig 1E). Trials that exceeded the mean ± 5 standard deviations (SD) were regarded as artifacts and thus excluded from further analysis [78]. For monkey 1, there were a total of 320 trials in the Flash condition and 216 trials in the Flash + AC condition, whereas for monkey 2, there were 295 trials in the Flash condition and 389 trials in the Flash + AC condition.

## Artifact rejection of electrical stimulation

AC artifact rejection was performed on the data after low-pass and notch filtering, but before segmentation, using MATLAB 2022b (MathWorks). ICA, a widely used method for decomposing mixed signals into independent sources, has been employed to remove the artifact induced by AC stimulation [85]. In this study, we applied the second-order blind

identification (SOBI) algorithm [86], a type of ICA embedded in the Fieldtrip toolbox [82], to remove the AC components oscillating at a sinusoidal frequency of 1.5 Hz [34]. After applying SOBI, sinusoidal components at 1.5 Hz were identified using the fast Fourier transformation (FFT). Although we also attempted to detect potential harmonics at 3 and 4.5 Hz, none were detected. This may be due to the direct measurement of signals in laminar recordings, which are minimally affected by volume conduction, unlike electroencephalogram signals recorded from the scalp [87]. Then, we employed a discrete Fourier transform filter to subtract the component from the data [88]. This approach, commonly used to eliminate sinusoidal powerline noise, allowed us to specifically remove the AC artifact while preserving neural signals. Finally, the cleaned components are back-projected to the contact level, resulting in LFPs free of AC artifacts. Power spectrum analysis was then conducted to evaluate the effectiveness of AC artifact rejection. As shown in Fig 1F, AC artifacts were successfully removed from the Flash + AC condition data.

## Biophysics of electrical stimulation

The data were analyzed using MATLAB 2022b (MathWorks) and the Fieldtrip toolbox. To extract electric potential oscillating at 1.5 Hz, the raw data was bandpass-filtered between 0.5 and 2 Hz using a fourth-order zero-phase, forward-reverse Butterworth filter, followed by downsampling to 1 kHz. Given that the signals were amplified by a factor of 10, we divided the electric potential by 10 to determine the actual amplitude of electrical stimulation. The electric fields were then calculated by applying the numerical gradient to the measured electric potential along contacts [12]. As the laminar probes were aligned radially with the cortical surface, this approach allowed us to quantify the electric field in the radial direction at each contact. Using FFT, we extracted the phase and amplitude of electric fields at each contact for the maximum frequency [11], which corresponded to the 1.5 Hz AC oscillation (Fig 1D). For each trial, the phase of AC at the onset of the visual stimulus was determined at each contact to identify the specific phase at which the visually evoked LFP occurred. This enabled us to assign a precise phase value of AC for each trial.

## Layer-specific LFP analysis

Laminar contacts were assigned to layers based on three approaches: (i) CSD analysis, (ii) electric field distribution, and (iii) anatomical references. CSD was computed by taking the second derivative of visual-evoked LFPs (filtered and segmented) along the contacts, using the data from the Flash condition [89]. This method is capable of capturing the early feedforward input in layer 4C during visual stimulation [35]. Based on the current sinks and sources in layer 4C [27] and anatomical references [15,25], the boundaries of layers 4AB and 5/6 were further determined based on distinct current source and sink distributions. However, pinpointing the precise location where layer 1 begins in the CSD distribution is challenging. Thus, we defined layer 1 as the region where the electric field began to increase, marking the boundary between the cortex and the outside of the cortex (Fig 4). This aligns with anatomical references, which estimate the depth of layer 1 to be approximately 0.1 mm [21]. Layers 2/3 were then identified as the region between the boundary of layer 1 and layer 4AB. Subsequently, contacts were categorized into five layers: four contacts were assigned to layer 4C and layer 5/6, two contacts to layer 4AB, five or six contacts to layer 2/3, and one or two contacts to layer 1 in both capuchin monkeys. We then obtained the layer-averaged visual-evoked LFPs for each trial in each layer. To assess LFP activity in white matter as a control condition, the laminar probe was further lowered. We lowered 1.3 mm from the base location to ensure that the five contacts from the tip of the probe were positioned within the white matter, rather than in the cortical layers. We then repeated the same analysis with the five contacts to obtain LFPs from the white matter.

## Phase dependency analysis

Phase dependency analysis was conducted using the data from the Flash + AC condition to determine whether LFP activity varies according to the phase of AC across cortical layers. Data were analyzed using MATLAB 2022b

(MathWorks) and CircStat toolbox [90]. We confirmed that visually evoked LFPs consistently appeared at 2.3 Hz in the time domain. Because the 2.3 Hz visual stimuli and 1.5 Hz AC stimulation were non-harmonic, flash stimuli naturally jittered across all phases of the electrical stimulation, resulting in a uniform phase distribution across trials (Fig G in S1 Text). In the previous step, the phase of AC was labeled for each trial. Using this information, layer-averaged trials were sorted into 20 phase bins, each spanning 18°. This approach is commonly used in phase-dependent analyses, as a higher phase resolution (smaller bin sizes) allows for finer detection of phase-dependent effects [91]. The LFP components, P1 and N1, were then determined for each phase bin, followed by averaging the trials to obtain the mean LFP component for each phase bin. Finally, we computed the mean direction and vector length of the phase-binned P1 and N1 components for each layer. The mean direction represents the preferred AC phase at which the LFP component tends to be strong, and the vector length quantifies how strongly LFP components preferentially align with the AC phase. Interestingly, we found a strong bimodal circular distribution of LFP components according to the AC phases in monkey 1, while monkey 2 showed a unimodal distribution. For instance, in monkey 1, the P1 component showed higher amplitudes at AC phases of 90° and −90° simultaneously, resulting in convergence in mean direction to 0°, and vector length to zero. To address this issue in monkey 1, we employed a different approach that accounts for bimodal circular data by using the *circ_axialmean* function embedded in the CircStat toolbox [92,93]. Additionally, we tested a larger bin size to increase the number of trials per bin and found that the results remained consistent, confirming the robustness of our findings (Fig H in S1 Text). In the Flash condition, where no electrical stimulation was delivered, we created a virtual 1.5 Hz sinusoidal signal to serve as a phase reference and performed the same analysis as above, enabling a comparison with the Flash + AC condition.

## Multi-unit activity analysis

To confirm that the in vivo LFP modulation and the cortical column modeling predictions were both associated with increased neural activity induced by AC, we additionally analyzed MUA recordings in monkey 2. The raw signals were separated into LFP (0.1–500 Hz) and MUA (300–5,000 Hz) ranges by analog filtering for both the Flash and Flash + AC conditions. To minimize any phase distortion, the MUA signals were further processed using a zero-phase digital band-pass filter (300–5,000 Hz). Spike detection was performed based on the method introduced in a previous study [94], where the detection threshold was defined as follows:

$$\text{Threshold} = \alpha \times \text{median} \left( \frac{|\mathbf{x}|}{0.6745} \right)$$

where *x* represents the filtered signals used for thresholding, and α is a factor for threshold levels (set to 3.5 in the current study), typically ranging between 2 and 4 [95]. This approach provides a robust estimate of noise levels and ensures consistent spike detection across different conditions. All events exceeding the threshold were classified as MUAs. The detected MUAs were segmented into epochs aligned with those used in the LFP analysis. For each trial, spike times were binned into 10 ms bins to obtain peristimulus time histograms relative to the visual stimulus onset [96]. The visual-evoked MUAs were then averaged across trials and across channels within each layer to obtain the layer-specific firing rates of MUAs. In addition, to assess phase-dependent modulation effects, we extracted trials from the Flash + AC condition where the visual stimulus onset was phase-locked to either the peak or the trough of the AC. While the phase-dependent LFP analysis utilized 20 phase bins to capture fine-grained changes, the MUA analysis focused only on trials aligned to the peak and trough phases because the peak and trough phases represent moments when neurons are most depolarized or hyperpolarized by the AC stimulation, making it easier to detect potential changes in firing activity. Specifically, 96 trials were phase-locked to the peak and 90 trials were phase-locked to the trough. The corresponding MUAs were separately analyzed to investigate phase-dependent modulation.

 

## V1 cortical circuit model

We utilized the biophysically detailed cortical column model of V1 established in the previous study [31]. This model incorporates both multicompartment and leaky-integrate-and-fire neurons, with a total of 230,924 neurons arranged in a cylindrical structure. It includes 17 neuronal classes distributed across cortical layers, with distinct excitatory and inhibitory populations. Thalamocortical input is provided by a lateral geniculate nucleus (LGN) module, while background activity is simulated using a Poisson process. This comprehensive model accurately represents the cortical circuitry and connectivity patterns of V1. The stimulus used in the simulation consisted of full-field flashes. Each trial comprised a 500 ms gray screen period, followed by a 50 ms white screen, and then a 350 ms gray screen. Ten trials were simulated.

The LFP evoked by the flash stimulus in simulations was derived from the extracellular potential using a fifth-order Butterworth low-pass filter with a cutoff frequency of 100 Hz, followed by a downsampling to 1 kHz and baseline correction (−50 to 0 ms). LGN spike trains used as input to the V1 circuit were generated with the FilterNet module provided with the model, using 17,400 'LGN units' [31]. The original model's synaptic arrangement was preserved: LGN-to-excitatory V1 neuron synapses were located on dendrites within 150 μm of the soma, while LGN-to-inhibitory V1 neuron synapses were placed on the soma and dendrites without distance constraints. The background input, firing at 1 kHz with a Poisson distribution, was not stimulus-dependent. Recurrent connection probabilities and synaptic placements from the original model remained unchanged in this study. Then, firing rates were calculated for each layer using a 2 ms time bin.

## Modeling AC stimulation

To integrate the electric fields into the V1 cortical circuit model, electric potentials were calculated for each neuronal segment based on the external electric field along the vertical axis of the model. These potentials were then set as the extracellular potentials within the 'extracellular mechanism' in the NEURON environment [97]. Given that an electric field is considered quasistatic, it can be divided into spatial and temporal components [98]. The spatial component of the electric field is modeled as uniformly distributed with strengths of 16 mV/mm, which both matches experimental values and aligns with previous simulation results [99]. The temporal component was modeled as a sinusoidal 1.5 Hz waveform. tACS was initiated at 250 ms and sustained for the remainder of the simulation. To investigate the phase dependency of the visual evoked response, the tACS waveform was shifted to align specific phases (peak or trough) with the flash onset at 500 ms. To assess the effects of LGN synaptic input on LFP, we recorded membrane currents in passive basal dendrites of 500 randomly selected excitatory neurons per layer for several key reasons. First, LGN synaptic inputs primarily target dendrites within 150 μm of the soma. Second, tACS-induced depolarization/hyperpolarization effects are substantially stronger in large, elongated excitatory neurons. Third, excitatory neurons constitute approximately 85% of the neuronal population in each layer, contributing more significantly to LFP. Measurements were taken for two phase conditions and a control condition (tACS with silent network). We summed currents across all segments and subtracted the control condition currents from the phase conditions to isolate LGN-induced effects from tACS-induced fluctuations.

## Statistical analysis

All statistical analyses were performed using MATLAB 2022b (MathWorks). For the comparison between the Flash and Flash+AC conditions, we employed a nonparametric cluster-based permutation test to assess whether electrical stimulation significantly improves visually evoked LFPs across contacts [82,100,101]. This approach is especially advantageous for time-series data analysis, as it provides robust findings without excessively conservative thresholds, such as those used in Bonferroni correction [100]. Concisely, trials from the Flash and Flash+AC conditions were compared using a $t$ test for each contact and time point and combined as a cluster according to their temporal adjacency. Cluster-level statistics were then calculated by taking the sum of the $t$-value within each cluster. We set the permutation iteration to 2,000 times and a critical value of 0.01 to strongly prevent false positives. Given that contacts were assigned to specific cortical layers, we can easily determine which layers exhibit a significant effect on LFP modulation by an external current.

A permutation test was also used to evaluate the significance of the change in the amplitude of P1 and N1 depending on AC phases [91]. For each component, the labeled phase values were randomly shuffled to generate surrogate datasets, followed by sorting them into phase bins and calculating the mean direction and vector length as in the previous step. Since the phase-binned surrogate data did not have bimodality, we employed a unimodal circular approach in the permutation test. Shuffling was repeated 5,000 times, resulting in 5,000 vector length values. The vector length of the original data was then standardized to a z-score using the mean and standard deviation of the surrogate distribution. A one-tailed p-value was calculated to determine the proportion of surrogate values smaller than the original data. If the original vector length exceeded 95% of the surrogate values, it was considered statistically significant. The same process was repeated for each layer. significant effect on LFP modulation by an external current.

To compare the firing rates of MUAs between superficial (layers 2/3) and deep layers (layers 5/6), we first averaged the firing rates across all time bins within each trial, resulting in a single firing rate value per trial. Paired t tests were then conducted within each condition (Flash or Flash+AC) to compare the firing rates between layers 2/3 and layers 5/6. To compare firing rates between conditions (Flash versus Flash+AC) within the same layer, unpaired two-sample t-tests were applied.

## Supporting information

**S1 Data.** Underlying data for Fig 5B and 5D.
(XLSX)

**S2 Data.** Underlying data for Fig F in S1 Text.
(XLSX)

**S1 Text. Supplementary Information.** Contains Figs A–I and Table A. **Fig A. CSD and LFP responses across layers. A)** Depth profile of the current source density (CSD) signals averaged across trials. The color yellow represents the current sink, while the color black represents the current source. **B)** LFPs recorded using multisite probe in monkey 2. Only the contacts involved in cortical layers are illustrated. Red lines represent visual-evoked LFPs in the Flash+AC condition, and blue lines depict LFPs in the Flash condition. **C)** Comparison of layer-averaged amplitudes of LFP components, P1 and N1, between the Flash (blue) and Flash+AC (red) conditions for both monkeys. **Fig B. Effects of AC on layer-specific visual-evoked LFPs in V1. A)** Monkey 1; **B)** Monkey 2. Normalized LFPs were averaged across the trials and contacts within each layer. Thick lines represent averaged LFP in the Flash condition (blue) and Flash+AC condition (red), with shades representing the standard deviation. **Fig C. Permutation test results for phase-dependent LFP modulation.** The permutation analysis for **A)** monkey 1 and **B)** monkey 2 corresponds to Fig 3. The gray histogram represents 5,000 permuted vector lengths. Gray dotted lines indicate the significance level. The blue and red lines indicate the unimodal vector length and bimodal vector length obtained from the original data, respectively. For monkey 1, the permutation test reveals significant bimodal phase preference in P1 and N1 with respect to the phase of AC within deeper layers (layers 4–6). For monkey 2, there is a unimodal phase preference in the amplitude of LFP components depending on the phase of AC in the deeper layers. **Fig D. Phase-dependent modulation of LFP components under virtual AC.** Amplitude of P1 and N1 components according to the phase of virtual AC in the Flash condition for both monkey 1 (A and B) and monkey 2 (C and D). **A, C)** The P1 and N1 components were sorted into 20 phase bins, followed by taking trial- and phase-averages for each layer. Gray thick and dotted lines represent the bimodal mean direction and unimodal mean direction of the amplitude of LFP components, respectively, based on the phase of virtual AC. **B, D)** Permutation test shows the absence of significant directional preferences in the LFP component with respect to the phase of AC across all cortical layers when virtual AC is assumed to be applied in the Flash condition. **Fig E. Effect of electrical stimulation on LFP activity in white matter. A)** Local field potentials (LFPs) in white matter in monkey 1. Normalized LFPs were averaged across trials and contacts within white matter. Thick lines and shades represent the averaged LFP and standard

deviation, respectively. Time indicates the duration from the flash visual stimulus onset. **B)** Circular distributions of the amplitude of LFP components, P1 and N1, depending on the phase of AC in white matter in monkey 1 (left column). The results from the permutation test are depicted for P1 and N1 components (right column). The permutation test shows that there is no significant directional preference in P1 and N1 amplitudes with respect to the phase of AC, regardless of whether the circular distribution was unimodal or bimodal (n.s., not significant). The gray histogram represents 5,000 permuted vector lengths. The blue lines and red lines indicate the unimodal vector length calculated from the original data and the bimodal vector length, respectively. **Fig F. Cortical column modeling results on layer-specific LFP and firing dynamics. A)** LFPs evoked by the flash stimulus in the cortical column model across layers. **B)** Comparison of LFP components (P1 and N1) obtained in vivo experiments and the firing rate between 50 ms and 150 ms. The P1 component does not correlate with an increase in the firing rate (left), while the N1 component shows a pattern that corresponds more strongly with the increasing firing rate as depth increases in both monkeys (right). **C)** Firing rate of LGN neurons, showing an initial spike around 50 ms (corresponding to the P1 period in simulations), followed by a second firing arising around the period of neural firing of V1 neurons. **D)** Normalized firing rate of V1 neurons when the visual stimulus was applied at the either peak or trough phase of AC. The firing rate is higher during the trough phase of AC than the peak phase in the deeper layers, while there is no comparable difference in firing rate between two phase conditions in the superficial layer. **Fig G. Uniformity of trial phase distribution under Flash + AC condition.** Polar histogram showing the number of trials across phase bins for the Flash + AC condition in both capuchin monkeys. The Rayleigh test confirmed a uniform phase distribution ($p > 0.05$). The average number of trials per phase bin was 12 for Monkey 1 and 21.67 for Monkey 2. **Fig H. Phase dependency of LFP components with wider phase bins.** Phase dependency analysis using a larger phase bin (30°). Amplitudes of the P1 and N1 components are shown according to AC phases, along with permutation test results. **(A, B)** P1 and N1 components for Monkey 1. **(C, D)** P1 and N1 components for Monkey 2. **Fig I. Modulation of P1 and N1 by AC phase relative to Flash baseline.** Phase-dependent modulation of P1 and N1 components under AC stimulation, compared to the Flash condition, in **(A)** monkey 1 and **(B)** monkey 2. Black lines indicate the average amplitude of the LFP components in the Flash condition. **Table A. Mean voltage and electric field for each layer in both monkeys.** (DOCX)

## Author contributions

**Conceptualization:** Sangjun Lee, Alexander Opitz.

**Data curation:** Gary Linn, Charles E. Schroeder, Arnaud Y. Falchier, Alexander Opitz.

**Formal analysis:** Sangjun Lee, Zhihe Zhao, Ivan Alekseichuk, Jimin Park, Sina Shirinpour.

**Funding acquisition:** Alexander Opitz.

**Investigation:** Sangjun Lee.

**Methodology:** Sangjun Lee.

**Supervision:** Alexander Opitz.

**Visualization:** Sangjun Lee.

**Writing – original draft:** Sangjun Lee.

**Writing – review & editing:** Sangjun Lee, Alexander Opitz.

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
