## [Editor Report · Decision Letter 0]

Dear Dr Lee,

Thank you for submitting your manuscript entitled "Layer-specific dynamics of local field potentials in monkey V1 during electrical stimulation" for consideration as a Research Article by PLOS Biology.

Your manuscript has now been evaluated by the PLOS Biology editorial staff as well as by an academic editor with relevant expertise and I am writing to let you know that we would like to send your submission out for external peer review.

Once your full submission is complete, your paper will undergo a series of checks in preparation for peer review. After your manuscript has passed the checks it will be sent out for review. To provide the metadata for your submission, please Login to Editorial Manager (https://www.editorialmanager.com/pbiology) within two working days, i.e. by Jan 10 2025 11:59PM.

Kind regards,

Christian

Christian Schnell, PhD

Senior Editor

PLOS Biology

cschnell@plos.org

---

## [Decision Letter · Decision Letter 1]

Dear Dr Lee,

Thank you for your patience while your manuscript "Layer-specific dynamics of local field potentials in monkey V1 during electrical stimulation" was peer-reviewed at PLOS Biology. It has now been evaluated by the PLOS Biology editors, an Academic Editor with relevant expertise, and by several independent reviewers.

In light of the reviews, which you will find at the end of this email, we would like to invite you to revise the work to thoroughly address the reviewers' reports.

As you will see below, the reviewers are overall supportive of your study, but raise a number of concerns. Reviewer 1 would like to see spike analyses as well and comments on the issue that the data are partially inconsistent between the two monkeys, with consequences for the modelling part in your manuscript. Reviewers 2 and 3 ask mostly for textual revisions and a few additional analyses.

Given the extent of revision needed, we cannot make a decision about publication until we have seen the revised manuscript and your response to the reviewers' comments. Your revised manuscript is likely to be sent for further evaluation by all or a subset of the reviewers.

**IMPORTANT - SUBMITTING YOUR REVISION**

*Re-submission Checklist*

*Published Peer Review*

*PLOS Data Policy*

*Blot and Gel Data Policy*

Sincerely,

Christian

Christian Schnell, PhD

Senior Editor

PLOS Biology

cschnell@plos.org

REVIEWS:

Reviewer #1: Lee et al. conducted a descriptive and exploratory study on the modulation of electrical stimulation on the local field potential (LFP) in primate V1. Their findings include: 1) the effects of electrical stimulation on visual-evoked LFPs (specifically the first positive peak, P1, and the first negative peak, N1, in the VEP/ERP) are layer-specific, with the strongest effects observed in the deep layers; 2) the effect of electrical stimulation is phase-specific for the P1 component but not for the N1 component in the ERP; and 3) a computational model based on rodent V1 can explain the layer-specific and phase-specific effects. While the experimental results are intriguing, the conclusions and explanations provided are relatively weak and require further support or evidence.

Major Concerns:

1. All experimental results in this study are based on LFP recordings across different V1 layers, without any supporting data or demonstration from spike activity in the corresponding layers. Although the laminar pattern of the LFP is interesting, the LFP is inherently uncertain due to volume conduction (see related comments in Minor Concerns 1 and 2). The authors should also present the effects on spike activity in their real data. Notably, the authors acknowledge the importance of spike activity, as they state in the discussion (page 11): "Additional recording modalities such as single-unit activity could further strengthen our results", I believe the authors have the opportunity to show results for multi-unit spike activity, as they mention in the methods (page 14): "remove possible contamination from multiunit spiking responses in the higher frequency range." Presenting experimental results from spike activity is particularly important since the modeling results in this study primarily focus on spike activity across different layers (Figure 5). Without experimental data on spike activity, it is challenging to connect the experimental findings with the modeling results.

2. Another concern is the inconsistency in the experimental results regarding phase-specific AC modulation on the P1 component in the ERP between the two monkeys, which is not well supported by the subsequent model simulation. While the study demonstrates that the P1 component is modulated by electrical stimulation in a phase-specific manner, the phase-specific modulation differs between the two monkeys. For monkey 1, the AC modulation is strong at two phase directions (opposite phases 180 degrees apart), whereas monkey 2 exhibits AC modulation with a single preferred phase direction. This inconsistency might be acceptable if the study merely aims to demonstrate phase-specific AC effects. However, it becomes a significant issue when the authors develop a computational model to explain this finding, as the modeling results show AC modulation with a single preferred phase direction, reproducing only the results for monkey 2 and not monkey 1. The authors should address the inconsistency in the experimental results and provide a more general modeling framework that accounts for the findings in both monkeys.

Minor Concerns:

1. The authors present laminar patterns for voltages and electric fields during AC stimulation in Figure 4 and contrast these results (first paragraph on page 8) with the voltage results for laminar patterns of the VEP/ERP (e.g., Figures 2 and 3). However, this comparison seems unfair. A more reasonable approach would be to show electric fields (the gradient of the voltage along the direction of the laminar probe) corresponding to those in Figures 2 and 3, and then compare voltage results for AC and ERP signals (or compare electric field results for the two signals).

2. In the second paragraph on page 10, the authors claim that their findings "highlights the anatomical and functional specificity of different cortical layers in their responsiveness to electrical stimulation" based on the observation that LFPs did not show modulation of evoked potentials by electrical stimulation in white matter (Fig S5). However, in the first paragraph on page 11, the authors state, "This effect is layer-specific, as the basal dendrites of large pyramidal neurons in the deeper layer show a high response to AC, which can spread to adjacent layer 4 due to volume conduction." If volume conduction can explain the results in layer 4, why does it not affect signals in white matter? The authors should present results for spike activity to address the uncertainty introduced by volume conduction.

3. While the study focuses on primate V1, many of the experimental citations are from rodent studies. For example, on page 3, the authors cite rodent studies to support their primate results: "A previous study demonstrated that flash-evoked LFPs increase with more depth in the rat brain in response to visual stimuli [29]" and "Cortical column modeling of mouse V1 has shown that neurons in layers 2/3 have the lowest firing rates, whilst those in layers 5/6 have the highest firing rates [30]." It would be preferable to include citations from monkey V1 studies. Additionally, since the computational model used in this work is based on rodent V1, the authors should explicitly address species differences in the discussion.

4. What is the significance of the findings? Why is the laminar pattern of electrical stimulation important? In the introduction, the authors state: "However, it is still unknown how electrical stimulation modulates evoked LFPs across cortical layers. While there is a comprehensive understanding of how the phase and amplitude of electrical stimulation affects neural dynamics at the level of single neurons [5-7],..." Although the authors provide several reasons for studying the effects of electrical stimulation on layer-specific LFPs, it is unclear how their results directly address these issues. The authors should explicitly state the significance of their findings in a more specific and concrete manner.

5. The authors state: "Two female capuchin monkeys (Cebus apella), each aged 15 years and weighing between 1.5 and 3kg, were included in the current study" The reported weights for 15-year-old capuchin monkeys seem low. The authors should double-check these values.

6. The description of the visual stimulus and electrical stimulation at the trial level is unclear. How long does each trial last? What are the start times for the visual stimulus and electrical stimulation within each trial? Figure 1 should include a demonstration of a real experimental trial to clarify these details.

7. In the last paragraph on page 8, the authors state: "occurred before neural firing (Figs 6A and S5A)." However, there is no Figure 6A. This should be corrected.

Reviewer #2: The authors of "Layer-specific dynamics of local field potentials in monkey V1 during electrical stimulation" describe the effects of transcranial low-frequency electrical stimulation of the occipital lobe on the local field potential across all layers of V1. With the application of a model of the cortical column, the authors show that phase-dependent changes in neurons can explain their in vivo results.

The scientific rigor employed in this work is commendable, most particularly in reference to the number of control experiments used to verify the authors' findings. However, the authors could put more effort into thoroughly explaining their methods and approach, and into situating their study in the broader literature of cortical stimulation. I recommend only minor revisions for this work.

In no particular order:

- Instead of referring to the experimental animals as "non-human primate (NHP)", or "monkeys", and only mentioning the species in the Methods, consider using the full name "capuchin monkey(s)" throughout.

- As both types of stimulation were applied at a fixed frequency (1.5 Hz and 2.3 Hz), there is the strong possibility of neural entrainment affecting the local field potentials observed. This possiblity needs to be addressed in the Discussion. In addition, the specific choice of anaesthetic (isoflurane) can suppress neural activity and impact entrainment, and also deserves to be addressed.

- While the choice of relative frequency between the AC and the visual flash was made such that the flashes would occur at different phases of AC, no specific numbers are given as to how many trials at each phase were completed, or why either 1.5 Hz or 2.3 Hz was specifically important. The introduction of a small jitter prior to each visual flash could also have prevented the possibility of neural entrainment to the visual stimuli.

- In Methods-Laminar recording, the authors state that they obtain LFPs by analog filtering between 0.1 and 500 Hz, then downsampling to 2 kHz. However, in Methods-Data preprocessing, the authors state that the LFPs were obtained by filtering the raw signal between 0.1 and 100 Hz, and downsampling the 1 kHz. Which statement is accurate?

- 216 trials of flash + AC were completed, and the authors state that trials were arranged into 20 phase bins of 18 degrees each. This leaves just over ten trials in each phase bin. What was the statistical power of the phase dependency analysis, and what motivated the choice of 20 phase bins of 18 degrees each, over, say, 10 phase bins of 36 degrees each, but with double the trials?

- Current-source density is a well-known approach capable of locating layer 4C, and the authors are clear with how they determine the upper border of layer 1. It is less clear how the authors were able to determine the borders between layers 1 and 2/3, layers 2/3 and 4AB, and layer 5 and 6, and more detail should be included about this approach. Because the results largely split into a difference between superficial and deep layers, it is not clear what is gained by attempting this level of granularity in layer separation without histology or robust, well-referenced and clear methodological approaches. The authors could also consider more recent techniques, such as the spike-phase index: https://doi.org/10.7554/eLife.84512, as another means of verifying contact placement, though this is not necessary.

- The choice of transcranial AC stimulation is not addressed in the text, and should be supported by a contextual introduction that includes reference to work using transcranial AC stimulation, and contrasted to other stimulation techniques in the Discussion. Brief mention is made of intracranial stimulation techniques (of which there are multiple works showing phase-dependency between stimulation event and the LFP: DOI:10.1523/JNEUROSCI.0928-18.2018 and 10.1088/1741-2552/ada828 come to mind) but not in how it might differ in its interactions with the LFP, and whether any potential differences might be worth exploring.

- There are quite a few minor typos and awkward grammatical constructions throughout the text, and the authors would benefit from a careful edit for readability. A few minor examples are included:

At least one reference (page 3, paragraph 3, line 2, reference 6) is in superscript, where others are inline in square brackets.

A typo in the Discussion (page 10, paragraph 3, line 5) "causing a more concentrated and increased electric field."

A typo in the Methods (page 14, paragraph 1, line 7) "the maximum electricl field in monkey 2 was lower than that in monkey 2"

Reviewer #3: Review of "Layer-Specific Dynamics of Local Field Potentials in Monkey V1 During Electrical Stimulation"

This study by Lee and collaborators examines how transcranial alternating current stimulation (tACS) modulates layer-specific local field potentials (LFPs) in the primary visual cortex (V1) of macaques. The researchers recorded laminar LFPs while presenting a 2.3 Hz flash visual stimulus, applying a 1.5 Hz sinusoidal current to the occipital lobe. Their key finding is that only the deeper cortical layers (layers 4-6) exhibit phase-dependent modulation of LFPs in response to AC stimulation. Using a computational cortical column model based on mouse V1, they suggest that these effects arise from phase-dependent changes in dendritic membrane currents.

The study employs a well-structured approach, integrating laminar recordings with computational modeling and a clear experimental design. While this study might contribute to our understanding of layer-specific neuromodulation, I would like the authors to address the following comments.

1. I might have missed it but the referencing scheme for LFPs was unclear to me. The authors state that current source density (CSD) analysis was used to identify cortical layers, but they do not clarify what referencing method was applied to the evoked LFPs. It is crucial to specify whether CSD, a bipolar, common average, or monopolar reference was used, as this impacts signal interpretation. In particular, recent studies showed that CSD approach and Bipolar referencing provided different output in terms of where the stronger (alpha) power is observed (Haegens et al. 2015)

2. As far as I understood the computational model is based on mouse V1, while the recordings are from macaques. Given known interspecies differences in laminar organization and synaptic processing, applying a mouse model to macaque data may lead to inaccurate extrapolations. This limitation could be addressed with a discussion on how species differences might impact their findings. I also struggled a bit in understanding the expected link between spikes and LFPs at different times.

3. The authors used independent component analysis (ICA) to remove AC-related artifacts. However, the absence of a visible 2.3 Hz peak in the power spectrum raises concerns. Since the flash stimulus was presented at 2.3 Hz, a corresponding peak should be evident. If this was omitted from the figures, the authors should clarify this and provide a justification.

4. If I understood correctly, the authors report that AC does not induce an overall increase in the P1 component but does exhibit a phase-dependent effect. This suggests that AC might increase P1 amplitude at some phases while decreasing it at others, leading to a net zero effect. Is that correct? If so, it should be discussed more explicitly.

5. The concept of "virtual AC frequency" was unclear to me. Did the authors generate a synthetic AC phase for control analyses, or were they attempting to detect intrinsic 1.5 Hz oscillations? A more precise explanation might be needed.

6. As LFPs primarily reflect dendritic currents, while spikes represent neuronal output. The study attempts to link P1 to thalamic input, which is reasonable given the role of early sensory processing. However, the relationship between LFP N1 and spiking activity is less clear to me. While the authors suggest a correlation, the mechanistic explanation remains ambiguous to me. Since spikes are the result of synaptic integration, whereas LFPs mostly reflect input and dendritic activity, it would be beneficial to further clarify how they establish this link, particularly in deeper layer.

7. Could the authors further discuss whether their observed effects would align with behavioral studies on AC stimulation (expected effect from deep layers' modulation).

References

Saskia Haegens et al. "Laminar Profile and Physiology of the alpha Rhythm in Primary Visual, Auditory, and Somatosensory Regions of Neocortex". In: The Journal of Neuroscience: The Official Journal of the Society for Neuroscience 35.42 (Oct. 2015), pp. 14341-14352. ISSN: 1529-2401. doi: 10.1523/JNEUROSCI.0600-15.2015.

---

## [Decision Letter · Decision Letter 2]

Dear Dr Lee,

Thank you for your patience while we considered your revised manuscript "Layer-specific dynamics of local field potentials in monkey V1 during electrical stimulation" for publication as a Research Article at PLOS Biology. This revised version of your manuscript has been evaluated by the PLOS Biology editors, the Academic Editor and one of the original reviewers.

Based on the reviews and on our Academic Editor's assessment of your revision, we are likely to accept this manuscript for publication, provided you satisfactorily address the following data and other policy-related requests:

* We would like to suggest a different title to improve its accessibility for our broad audience: "Layers of the monkey visual cortex are selectively modulated during electrical stimulation"

* Please add the links to the funding agencies in the Financial Disclosure statement in the manuscript details.

* Please include the approval/license number of the ethical approval for the animal experiments.

* DATA POLICY:

Regardless of the method selected, please ensure that you provide the individual numerical values that underlie the summary data displayed in the following figure panels as they are essential for readers to assess your analysis and to reproduce it: 5BD and S6D.

* CODE POLICY

* Please note that per journal policy, the model system/species studied should be clearly stated in the abstract of your manuscript.

We expect to receive your revised manuscript within two weeks.

*Published Peer Review History*

*Press*

Sincerely,

Christian

Christian Schnell, PhD

Senior Editor

cschnell@plos.org

PLOS Biology

Reviewer remarks:

Reviewer #1: The authors have addressed my concerns and I have no further comments.

---

## [Editor Report · Decision Letter 3]

Dear Sangjun,

Thank you for the submission of your revised Research Article "Layers of the monkey visual cortex are selectively modulated during electrical stimulation" for publication in PLOS Biology. On behalf of my colleagues and the Academic Editor, Christopher Pack, I am pleased to say that we can in principle accept your manuscript for publication, provided you address any remaining formatting and reporting issues. These will be detailed in an email you should receive within 2-3 business days from our colleagues in the journal operations team; no action is required from you until then. Please note that we will not be able to formally accept your manuscript and schedule it for publication until you have completed any requested changes.

PRESS

We frequently collaborate with press offices. If your institution or institutions have a press office, please notify them about your upcoming paper at this point, to enable them to help maximize its impact. If the press office is planning to promote your findings, we would be grateful if they could coordinate with biologypress@plos.org. If you have previously opted in to the early version process, we ask that you notify us immediately of any press plans so that we may opt out on your behalf.

Sincerely, 

Christian

Christian Schnell, PhD

Senior Editor

PLOS Biology

cschnell@plos.org